# SGAM: Building a Virtual 3D World through Simultaneous Generation and Mapping

**Yuan Shen**[1]    **Wei-Chiu Ma**[2]    **Shenlong Wang**[1]
[1]University of Illinois at Urbana-Champaign    [2]Massachusetts Institute of Technology
{yshen47, shenlong}@illinois.edu
weichium@mit.edu

## Abstract

We present simultaneous generation and mapping (SGAM), a novel 3D scene generation algorithm. Our goal is to produce a realistic, globally consistent 3D world on a large scale. Achieving this goal is challenging and goes beyond the capacities of existing 3D generation or video generation approaches, which fail to scale up to create large, globally consistent 3D scene structures. Towards tackling the challenges, we take a hybrid approach that integrates generative sensor modeling with 3D reconstruction. Our proposed approach is an autoregressive generative framework that simultaneously generates sensor data at novel viewpoints and builds a 3D map at each timestamp. Given an arbitrary camera trajectory, our method repeatedly applies this generation-and-mapping process for thousands of steps, allowing us to create a gigantic virtual world. Our model can be trained from RGB-D sequences without having access to the complete 3D scene structure. The generated scenes are readily compatible with various interactive environments and rendering engines. Experiments on CLEVER and GoogleEarth datasets demonstrates ours can generate consistent, realistic, and geometrically-plausible scenes that compare favorably to existing view synthesis methods. Our project page is available at https://yshen47.github.io/sgam/.

## 1 Introduction

Human perception goes way beyond simple recognition and reconstruction. Our extraordinary abilities not only allow us to make sense of what we see, but also enable us to *imagine* what we do not (*e.g.*, reason what the scene looks like outside the image). With a simple glance, we can effortlessly re-build a mental world, that may not be exactly like the original, but is perceived by our brain to be the same. In fact, it is such a profound ability drawing us apart from existing AI systems.

The goal of this paper is to equip computational machines with similar capabilities. We aim to develop a program that can create an arbitrarily large, realistic, and consistent *3D world* from *a single snapshot of the scene*. The task is of paramount interest to many applications in computer vision (60; 40; 74), computer graphics (65; 65), geography (5), and robotics (13; 44), since it unlocks numerous potentials. For instance, it may allow us to build an interactive virtual environment without any costly and laborious 3D modeling pipeline (12).

Indeed, there has been a consistent pursuit within the computer vision and graphics community in the past few decades (17; 16; 85; 65; 9; 68; 72), where people attempt to design algorithms and models that are capable of extrapolating and hallucinating the world from a single input image. Unfortunately, a large body of efforts have been focusing on modeling in the 2D image space (17; 16; 85; 18; 37; 9), or require external 3D assets and manually specified rules for restricted 3D generation (10). Recently, with the advent of deep generative models (36; 76; 40; 61; 60), researchers have made great strides in unconstrained 3D syntheses. However, due to the highly complex structure of the task, these approaches mostly focus on object-centric scenarios, or generating small-scale environments.

36th Conference on Neural Information Processing Systems (NeurIPS 2022).

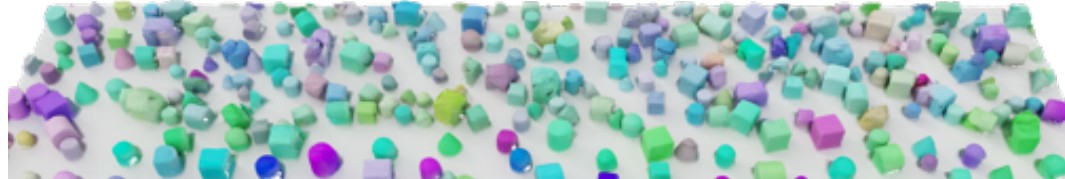

Figure 1: **A large-scale virtual world created after 2000 SGAM steps**. SGAM is an auto-regressive generative framework that simultaneously generates sensor data at novel views and builds a 3D map.

With these problems in mind, we present Simultaneous Generation and Mapping (SGAM), a novel 3D scene generation algorithm. SGAM builds upon insights from state-space representation in SLAM (48; 23) as well as recent generative models (73; 18; 66). At its core lies two key modules: (i) *a generative sensor module* that takes a scene representation and a novel query viewpoint as input and renders a photo-realistic RGB-D image of a 3D scene; and (ii) *a mapping module* that exploits the newly generated RGB-D observation at a given camera pose and updates the scene representation. Given an arbitrary camera trajectory, our method iteratively applies this procedure to construct the 3D scene. By grounding scene generation with "a map", we can produce a realistic and spatially consistent 3D world at a significantly larger scale than any previous 3D generation effort without drifting or collapsing. Additionally, our generated scenes can be represented explicitly as textured 3D models, making them readily compatible with various interactive environments and rendering engines. We note that we are not the first to tackle the task of scene synthesis under large viewpoint differences (56; 60; 40). Several recent efforts have demonstrated promising results in this direction, from which our method is inspired. Nonetheless, these works do not produce a consistent 3D scene representation of the infinite world and potentially suffer from mode collapse (60; 40).

We validate the efficacy of SGAM on two large-scale 3D scene datasets, Clevr-infinite and GoogleEarth. Both benchmarks produce substantially larger 3D scenes than existing datasets. This allows us to benchmark large-scale 3D scene generation. We evaluate the standard metrics for perceptual image quality, such as PSNR, SSIM (75), LPIPS (82), IS (63), and FID (27). We further benchmark the realism of the generated 3D scenes in terms of Jensen-Shannon divergence and maximum-mean discrepancy (22). Experimental results suggest that 1) our method produces more realistic results than existing single-image novel view synthesis methods; 2) our approach generates a more meaningful 3D world than the prior perpetual view generation algorithms.

## 2 Related Work

**Image generation:** How to synthesize an image realistically has been a long-standing problem in computer vision. The task dates back to 60s (32; 4) where researchers attempted to generate textures by matching statistics (42; 26). Through parametric sampling (85) or non-parametric matching (17; 16), they were able to synthesize an infinite amount of high-fidelity texture images. Unfortunately, these approaches fall short when applied to natural images, since the images have much higher complexity. Recently, with the help of deep generative models (21; 35; 55; 28; 66), researchers have demonstrated promising results on generating photo-realistic images (30; 33; 34). With proper design and inductive biases (73; 18; 11), they are even able to scale the output to mega-pixel level (70; 9). In this paper, we build our generation module on top of VQ-GAN (18). However, instead of treating scene generation as a pure 2D task as in the past, we explicitly consider the relationship between appearance and geometry — both at the input level and the output level. By encoding both information into the quantized codebook, we are able to "grow" the 3D scene from an initial seed image and generate a boundless world, which is not exactly like the original, but will be perceived by humans to be the same region.

**3D Generation:** 3D modeling and synthesis have been an active yet challenging research problem for decades (8; 6; 68). Recently, with the development of image generation techniques (21; 35; 57), the field has experienced a rapid growth (78; 49; 69). Drawing inspiration from its corresponding 2D analogue, researchers have been able to generate high quality point clouds (38; 79; 7), voxels (19; 77), meshes (24; 20), etc. Unfortunately, since 3D solution space is much more intricate than that of 2D, these approaches are typically object-centric; also, they mostly focus on generating common

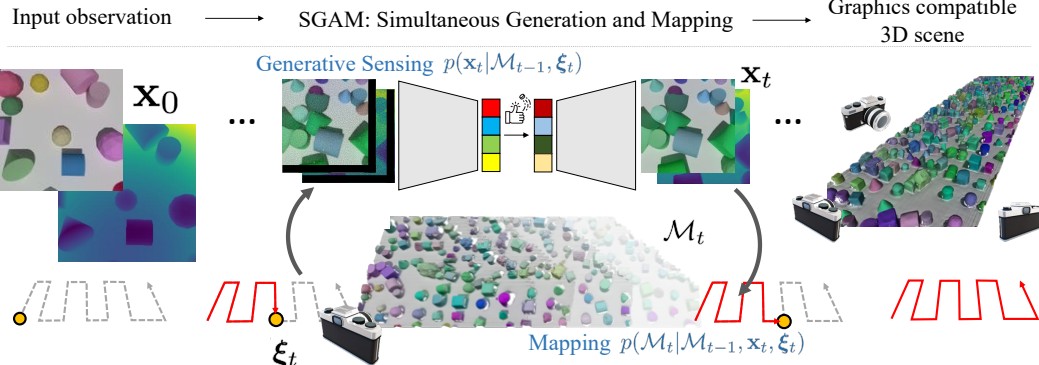

Figure 2: **Overview of our approach.** Our method consists two steps. 1) a generative sensing step $p_\theta(\mathbf{x}_t|\mathcal{M}_{t-1}, \boldsymbol{\xi}_t)$ which renders a realistic and complete sensor observation $\mathbf{x}_t$ given an (incomplete) scene representation $\mathcal{M}_t$ and a query viewpoint $\boldsymbol{\xi}_t$; 2) a mapping module $p_\theta(\mathcal{M}_t|\mathcal{M}_{t-1}, \mathbf{x}_t, \boldsymbol{\xi}_t)$, which integrates the new sensory observation to update the scene. Given an initial snapshot and a camera trajectory, we repeatedly run this generation-mapping process to create the virtual 3D world.

objects in our daily lives (25; 39; 14; 15). To enable scene-level synthesis, researchers have sought to incorporate more structures into the generation pipeline (47; 54; 51; 53), such as reducing the output space from full 3D to predefined, compact representation (52; 46; 67). While these strategies greatly alleviate the issue, the generated scene scale is still rather limited (*e.g.*, an indoor environment). In this work, we push the boundary of 3D generation systems and present a model that is capable of generating coherent 3D scene structure over thousands of meters. The produced high-quality 3D world not only allows us to build a large-scale map, but also provides geometric grounding for visual appearance synthesis.

**Generative sensor models:**     Generative sensor models serve critical roles in graphics/robotics simulation systems (44; 10). Their goal is to faithfully simulate the sensor measurement in real world. Depending on the sensor modalities (*e.g.*, rgb/depth cameras, LiDAR scanner), the output can be either photometric or geometric quantities. These methods usually assume an underlying (proxy) 3D geometry is given (80; 59) and then perform conditional generation (43). The geometry can take various forms (*e.g.*, point clouds (2), surfels (80; 44), meshes (58), etc), as long as it can effectively ground the system. Our full model is a combination of a generative sensor module and a mapping module. The former takes as input RGBD images and simulates both the appearances and the geometry of the scene.

**Single image novel view synthesis (NVS):**     Our work is also related to single image NVS approaches (62; 71; 76; 29), whose goal is to simulate novel views of a scene using only visual cues from one input image. These methods typically rely on neural networks to learn statistical priors from data. Together with carefully designed scene representations (*e.g.*, radiance fields (81; 74), multi-plane images (71), layered depth images (72; 64), etc), they are able to synthesize scene appearance with mild viewpoint change. To push the frontier even further, more recently, researchers have proposed to go beyond local view synthesis, and instead simulate the scene under larger viewpoint change (60; 40; 36; 61; 56). For instance, generate one room from the other. The task is extremely challenging as it requires the model to learn not only the underlying scene representations, but also the long-range dependencies. Without proper state/map representations, the generated output will easily drift away and the constructed world will no longer be coherent (40; 61). To address this issue, in this work, we develop a model that simultaneously generates novel views and builds a map. By integrating synthesized results into the map, we are able to query relevant information whenever we re-visit a spot and then decode, allowing us to produce realistic and consistent video footage.

## 3   SGAM: Simultaneous Generation and Mapping

In this paper, we seek to devise a method that can generate an arbitrarily large, realistic, and consistent *3D world* from *a single snapshot of the scene*. Based on the observation that current 3D scene generation approaches are prone to drifting and often produce inconsistent 3D worlds, we present

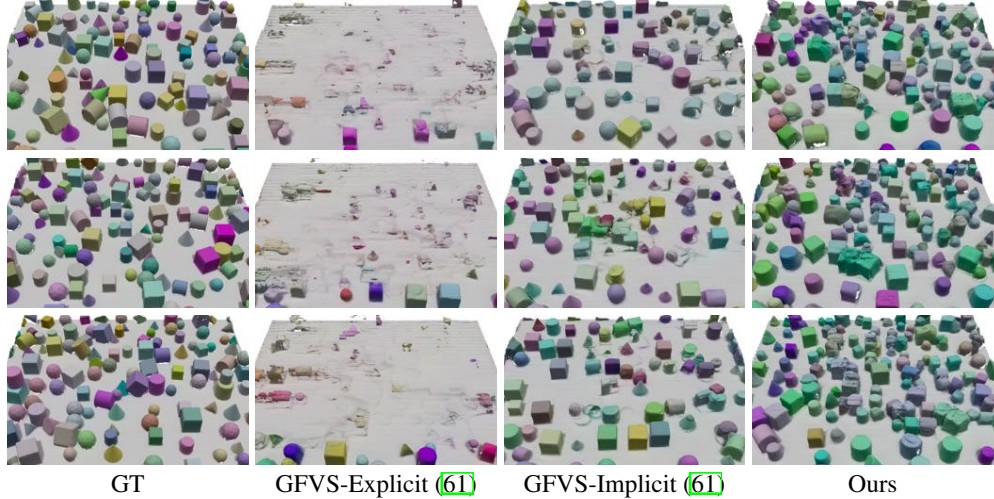

GT      GFVS-Explicit [61]      GFVS-Implicit [61]      Ours

Figure 3: **Randomly sampled scenes from CleverInfinite dataset.**

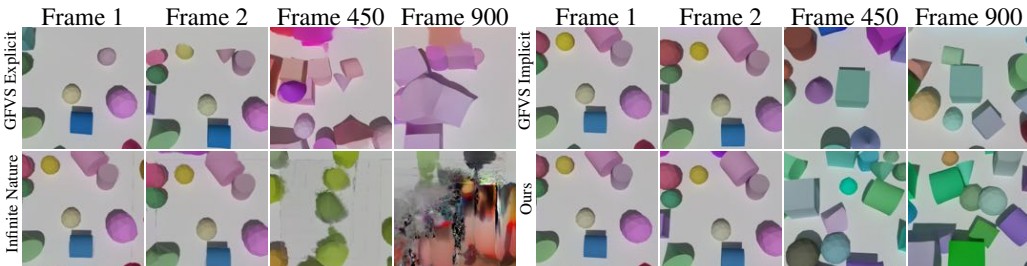

Figure 4: **Unrolling results on CLVER-Infinite.**

a novel framework, dubbed as SGAM, that simultaneously generates novel scenes and constructs world maps. By grounding the generation and the mapping process in a tightly coupled fashion, we are able to produce a 3D world at a significantly larger scale than any previous 3D generation effort.

We unfold this section by formulating our task in a probabilistic framework. Then we describe how each module can be implemented with a neural network and how we conduct inference to construct the virtual 3D world. Finally, we discuss the learning procedure, our design choices, and the relationships to existing methods.

## 3.1 Problem Formulation

Given a camera trajectory $\{\boldsymbol{\xi}_t \in \mathbb{SE}(3)\}$ over discrete time steps $\{t\}$ and an initial RGB-D sensor observation $\mathbf{x}_0 \in \mathbb{R}^{W \times H \times 4}$, SGAM aims to compute an estimate of the sensor observation $\mathbf{x}_t \in \mathbb{R}^{W \times H \times 4}$ and a perceived map of the environment $\mathcal{M}_t$ as a set of 3D surface points and associated colors $\{\mathbf{p}_i \in \mathbb{R}^3, \mathbf{c}_i \in \mathbb{R}^3\}_{1 \ldots N_t}$. Due to the inherent randomness, we consider all the quantities to be random variables, so the objective is to model:

$$p(\mathcal{M}_{1 \ldots t}, \mathbf{x}_{1 \ldots t} | \boldsymbol{\xi}_{1 \ldots t}) \tag{1}$$

Using Bayes rule and Markov property, SGAM factors the joint probability into a product of conditional probability over all steps and concerns the following auto-regressive model at each step $t$:

$$p(\mathcal{M}_t, \mathbf{x}_t | \mathcal{M}_{t-1}, \boldsymbol{\xi}_t) = p(\mathbf{x}_t | \mathcal{M}_{t-1}, \boldsymbol{\xi}_t) p(\mathcal{M}_t | \mathcal{M}_{t-1}, \mathbf{x}_t, \boldsymbol{\xi}_t) \tag{2}$$

At each step SGAM needs to estimate two conditional probabilities: 1) a generative sensor module $p_\theta(\mathbf{x}_t | \mathcal{M}_{t-1}, \boldsymbol{\xi}_t)$ rendering a realistic and complete sensor observation $\mathbf{x}_t$ given an (incomplete) scene $\mathcal{M}_t$ and a query viewpoint $\boldsymbol{\xi}_t$; 2) a mapping module $p_\theta(\mathcal{M}_t | \mathcal{M}_{t-1}, \mathbf{x}_t, \boldsymbol{\xi}_t)$, which integrates the new observation to update the scene. Next we will describe each module in detail.

| Renderer | PSNR↑ | SSIM↑ | LPIPS↓ | ref-FID↓ | FID↓ | Runtime ↓ |
|---|---|---|---|---|---|---|
| GFVS-implicit [61] | 23.06 | 0.872 | 0.138 | 10.47 | 22.85 ±1.66 | 33.65 |
| GFVS-explicit [61] | 18.53 | 0.812 | 0.256 | 15.66 | 23.90 ±1.02 | 10.86 |
| InfiniteNature [40] | 20.48 | 0.876 | 0.247 | 21.09 | 33.05 ±1.81 | 0.15 |
| Ours | **24.65** | **0.901** | **0.111** | **7.80** | **21.91** ±1.28 | 0.26 |

Table 1: **Comparison study** of ours and baseline methods on the CLEVR-Infinite validation set. The unit for runtime is average second per image. FID is computed over 2500 samples for each method.

## 3.2 Generative Sensing

We first describe our generative sensing module $p(\mathbf{x}_t|\mathcal{M}_{t-1}, \boldsymbol{\xi}_t)$, which samples a new RGBD observation $\mathbf{x}_t$ given a query viewpoint $\boldsymbol{\xi}_t$ and the current scene $\mathcal{M}_{t-1}$. Three desired properties are 1) *consistency* the generated sensor is consistent with existing 3D scene; 2) *realism*: the resulting images look realistic, regardless of how many auto-regressive steps we run; 3) *completeness*: the sensing module should produce complete observation over both explored and unexplored areas, allowing our scene to expand as camera moves. Towards these goals, we propose a two-step process including neural rendering and conditional generative refinement as shown in Figure 2.

**Scene rendering:**  We firstly create a guidance image at the target pose by rendering the current scene $\mathbf{x}'_t = \pi(\mathcal{M}_{t-1}, \boldsymbol{\xi}_t)$, where $\pi$ is an image-based rendering procedure. Given a triangular mesh extracted from the scene representation $\mathcal{M}_{t-1}$, the image-based rendering pipeline first produces a dense depth map at the target pose $\boldsymbol{\xi}_t$ through rasterization. The rendered depth map is then used for inverse warping, producing an RGB image by transferring color appearance from existing sensor observations to the target image. A geometry consistency check will be conducted between each pixel at the rendered target depth and source depth, ensuring the color comes from the same point in 3D. The image-based rendering step gives us an incomplete RGBD observation, where the missing pixels are due to the incompleteness of the current scene representation. Nevertheless, this step promotes the consistency between the generated sensory data and our current knowledge about the scene, built upon the past generated sensory data.

**Generative refinement:**  Next, we generate the final complete virtual sensor observation $\mathbf{x}_t$ based on the incomplete rendering $\mathbf{x}'_t$. First, the incomplete image $\mathbf{x}'_t$ is fed into an encoder network, $f_{enc}$ which outputs a continuous-valued latent feature map $\hat{\mathbf{z}}$. Then, we quantize $\mathbf{z}$ with a pre-trained codebook $\mathbf{D}$, by sampling the closest quantized feature $\mathbf{z}_q$:

$$\mathbf{z}_q = q(\hat{\mathbf{z}}, \mathbf{D}) := \left( \underset{\mathbf{z}_k \in \mathbf{D}}{\operatorname{argmin}} \|\hat{\mathbf{z}}_{ij} - \mathbf{z}_k\| \right) \in \mathbb{R}^{h \times w \times n_z}, \text{ where } \hat{\mathbf{z}} = f_{enc}(\mathbf{x}'_t) \tag{3}$$

where, $h$ and $w$ is the height and width of latent feature map, and $n_z$ is the embedding dimension.

Alternatively for $q(\hat{\mathbf{z}}, \mathbf{D})$, to encourage diversity, we could also sample each quantized embedding from the following distribution of its corresponding latent code $\hat{\mathbf{z}}_{ij}$ by measuring the soft-max similarity to each code word embedding $\mathbf{z}_k$ in the pre-trained codebook $\mathbf{D}$. We provide qualitative results of temperature sampling on CLEVR-Infinite dataset in our supplementary material. Mathematically,

$$q(k = r|\mathbf{x}'_t) = \frac{\exp(-d_r/T)}{\sum_j \exp(-d_j/T)}, d_k = \|\mathbf{z}_k - \hat{\mathbf{z}}_{ij}\|_2^2$$

where $q(k = r|\mathbf{x}'_t)$ is the probability of $r$-th latent code following a K-dimensional categorical distribution; $K$ is the size of the codebook; $T$ is a temperature parameter; and $d_k$ is the quantization error between the feature vector and the codebook element.

Finally, we send $\mathbf{z}_q$ to a decoder network $f_{dec}$ to get complete observation $\mathbf{x}_t$. Our generative sensing model is efficient due to our one-shot denoising procedure during the sampling stage, as opposed to the computationally-expensive sequential decoding in other VQGAN-based methods ([18; 61]).

## 3.3 Mapping

**Representation**  Maintaining an efficient and flexible scene representation is crucial for high fidelity 3D world generation at a large scale. Inspired by its success in depth-based SLAM ([48]) and 3D

| | Image metrics | | Scene metrics | | | | |
|---|---|---|---|---|---|---|---|
| Model | FID ↓ | IS ↓ | JSD $(10^{-2})$ ↓ | MMD $(10^{-5})$ ↓ | MinDist ↓ | 1-NNA ↓ | Cov ↑ |
| GFVS-implicit [61] | **16.14** | 1.56 ±0.05 | 0.775 | 4.510 | **0.205** | 0.675 | 0.376 |
| GFVS-explicit [61] | 82.82 | 3.11 ±0.18 | 10.870 | 211.500 | 0.215 | 0.825 | 0.122 |
| Ours | 26.60 | **1.55** ±0.08 | **0.656** | **4.441** | 0.221 | **0.585** | **0.454** |

Table 2: **Scene-level comparison study** on unrolling 30x30 scenes with prior arts on CLEVR-Infinite. For scene-level metrics, we use 3D hist on $15^3$ resolution with Cosine distance.

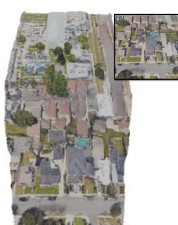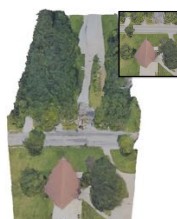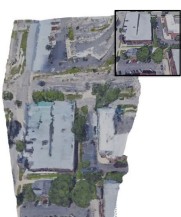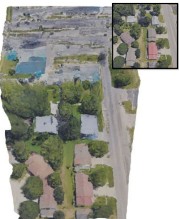

Figure 5: **Generated 3D map on GoogleEarth-Infinite dataset**. Each sequence unrolls for 70 steps with the same relative movements between frames. The top right is the initial RGB image.

reconstruction [83], we leverage hashing based volumetric representation [50] for $\mathcal{M}_t$. Each voxel **p** in the space maintains a signed distance value $\mathcal{M}_t(\mathbf{p}) : \mathbb{R}^3 \to \mathbb{R}$, where the magnitude encodes the distance to the surface boundary, and the sign represents whether it is interior or exterior. Such representation is flexible, allowing us to update online at each step when new sensory data is generated. Additionally, voxel-hashing enables sparse storage, allowing the map to scale to huge scenes.

**Integration**     At each time $t$, given a new observation $\mathbf{x}_t$, we update our map $\mathcal{M}_t$ following KinectFusion [48]. Specifically, a weighted sum update is conducted for points along the camera ray:

$$\mathcal{M}_t(\mathbf{p}) = \frac{w_{t-1}(\mathbf{p})\mathcal{M}_{t-1}(\mathbf{p}) + O(\mathbf{x}_t, \mathbf{p})}{w_{t-1}(\mathbf{p}) + 1}; w_t(\mathbf{p}) = w_{t-1}(\mathbf{p}) + 1$$

where $O(\mathbf{x}_t, \mathbf{p})$ is the observed signed distance function at **p**, calculated by the new sensor observation $\mathbf{x}_t$; $w_t$ is a counter that summarizes how many steps that **p** has been updated. After each integration step, we will also conduct marching cube [41] to extract triangular mesh, allowing us to render images from a new viewpoint during the generative sensing process.

### 3.4   Learning

SGAM adopts a two-stage training pipeline. In the first stage, inspired by VQGAN [18], we first pre-trained a discrete-latent encoder, decoder, and codebook on the clean RGB-D image dataset in an auto-encoding fashion. During training, we perform online re-initialization of the codebook embedding by performing k-means clustering over a past feature cache on the unquantized feature embedding space, whenever the active codeword hit rate within a fixed time interval drops below a pre-defined threshold. Following the practice in GFVS [61], we use $L_1$ reconstruction loss, perceptual loss [82], patch-gan-based discriminator loss [30] and commitment loss [18].

Our second stage aims to train the encoder so that it learns to complete the latent code space. Specifically, we fine-tune the encoder with incomplete/complete image pairs. The incomplete RGB-D images are simulated through inverse warping from neighboring viewpoints. We use perceptual loss [82] and patch-gan-based discriminator loss [30] at this stage. In this stage, we freeze the codebook and decoder parameters. We find this mechanism helps maintain the output images realistic even after many unrolling steps during inference.

We use Adam to train our entire network. Learning through a non-differentiable quantization step is hard. To stabilize training, we first train the encoder-decoder network without quantization for the first 30k iterations and then add it back and train the entire network for another 60k iterations using the straight-through gradient estimator trick [3] to back-propagate gradient.

## 4   Experiments

We first evaluate the effectiveness of SGAM on a large-scale synthetic dataset. Next, we comprehensively study the characteristics of our method. Finally we showcase how SGAM can be applied in real-world scenarios to generate high-quality 3D worlds.

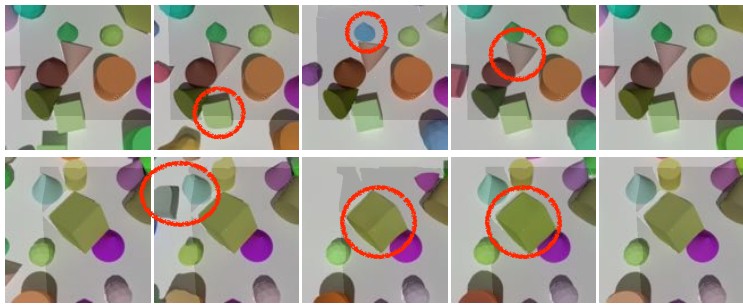

Ground-truth InfiniteNature (40) GFVS-Explicit (61) GFVS-Implicit (61) Ours

Figure 6: **Qualitative result of one-step predictions on CLEVR-Infinite.** Shaded areas are visible regions in source views based on forward warping.

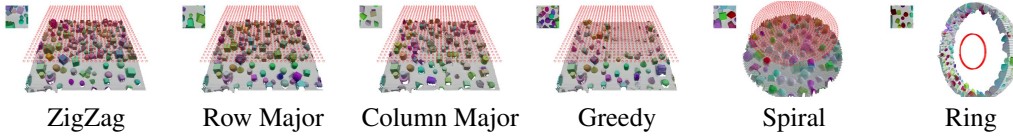

ZigZag          Row Major          Column Major          Greedy          Spiral          Ring

Figure 7: **Diversity samples of the 3D scene using different virtual camera trajectories.**

## 4.1 Setup

**Dataset:** Existing static, large-scale 3D datasets, such as ACID (40) and RealEstate10K (84), do not provide accurate 3D world for evaluation. We thus exploit Blender and the assets from CLEVR (31) to render an extremely large-scale synthetic benchmark, which we called CLEVR-Infinite. CLEVR-Infinite contains 72 training scenes and 8 test scenes. For each scene, we sampled a fixed number of objects from the asset bank and randomly placed them on the ground plane. The asset bank consists of four primitive shapes with random scales, materials and colors. The cameras are distributed uniformly among the $50 \times 50$ grid plane right above and parallel to the ground. We set the pitch (*i.e.*, the angle between the look-at vector and the plane) of the cameras to be $80°$. In total, the dataset contains 200k RGB-D images at the resolution of $256 \times 256$.

**Metrics:** We evaluate the generated 3D world both *locally at the image level* and *globally at the scene level*. The former measures to what degree the synthetic scene reproduces the local details, while the latter quantifies how well the virtual world matches the statistics of the original counterpart.

Following previous work (45; 81), we first use Peak Signal-to-Noise Ratio (PSNR), Structural Similarity (SSIM), and LPIPS (82) to measure the image level similarity with respect to the GT. However, since our model is generative, we might synthesize scenes that are plausible yet different from the original one. We thus further employ Fréchet inception distance (FID) (27) to measure the generative quality. Specifically, two FID score are reported: (i) ref-FID, which we compare against the GT; and (ii) FID, which we compare against a separate set of images from test set.

As for scene-level evaluation, we follow prior art in 3D generative models (1; 79) to gauge the overall geometry quality with a diverse set of metrics: Jensen-Shannon divergence (JSD), maximum mean discrepancy (MMD), coverage (COV), minimum matching distance (MinDist), and 1-nearest neighbor accuracy (1-NNA). Specifically, we first divide the whole scene into non-overlapping regions. Each region is then discretized into $15^3$ voxels. Finally, we compute the 3D histogram within each local grid and employ cosine distance to compute the metrics. To evaluate the realism, we also compute the Inception Score (IS) and FID against random sampled scenes from the test set.

We report the runtime of a single prediction step, benchmarked on Nvidia Quadro 8000.

**Baselines:** We compare SGAM against state-of-the-art approaches in long range scene generation: GFVS (61) and InfiniteNature (40). For GFVS, we consider both the explicit-image and implicit-depth variants with the following modifications: first, we use a single codebook to encode RGB-D rather than two, which better combines texture and geometry during scene expansion; second, to support multi-image input, we conduct average pooling over the features of source images before passing it to an MLP to infer the target code. For InfiniteNature, since the authors did not release

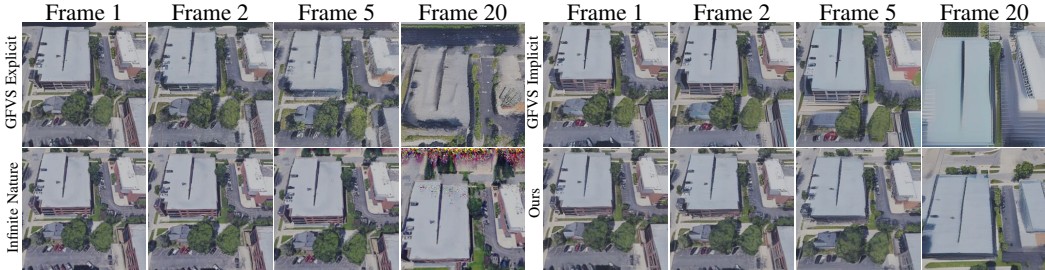

Figure 8: **Unrolling results on Google-Earth.**

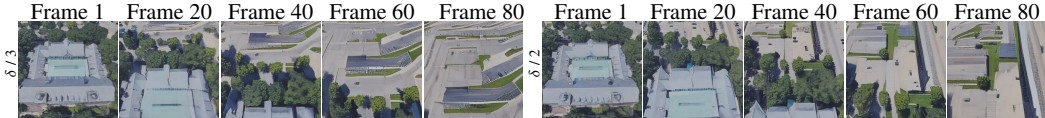

Figure 9: **Diversity samples of the 3D scene using different step size.**

their training scripts, we re-implement their method in Pytorch. We remove the geometric grounding step since we find it harmful when we unroll the scene in CLEVR-Infinite.

**Implementation details:** During full scene evaluation, we generate the scene in a zig-zag fashion on a $30 \times 30$ grid. We start from coordinate $(0, 0)$ and unroll for 900 steps. The above procedure is repeated for 10 times with different seed images. For a fair comparison, GFVS-implicit, GFVS-explicit and SGAM share the same codebook embedding and the same weights for decoder. We set the vocabulary size to 16384. We implement SGAM in PyTorch, and train with a batch size of 8 on 2 Nvidia A40 GPU until convergence. For other baselines, we maximize batch size to fit GPU memory, and train until convergence. During the first-stage codebook training, we follow the exact training config as GFVS first-stage training in the official codebase. GFVS-implicit, GFVS-explicit and SGAM share the same learning rate at 0.0625 for the second-stage trainig.

## 4.2 Experimental results

As shown in Tab. 1, SGAM achieves superior or comparable performance across all image level metrics. Comparing to the second best approach, GFVS-Implicit, we are much more efficient. We use less CPU and GPU memory and our runtime is 150 times faster. Fig. 6 shows a few one step prediction results. SGAM is able to faithfully reconstruct the color and the shape of the assets within the visible area, and generate the invisible region in a coherent, harmonic fashion. When unrolling multiple steps (see. Fig. 4 and Tab. 2), baselines such as InfiniteNature and GFVS-explicit suffer from error accumulation, and collapse catastrophically. GFVS-implicit generates objects that resemble the GT data, yet it sacrifices the speed and fails to enforce consistency across frames (see Fig. 6). Our approach, in contrast, is able to produce consistent and realistic results in an efficient manner.

## 4.3 Analysis

**Importance of two-stage training:** Two-stage training is critical. Removing either stage degrades our performance. Without the first-stage codebook training, the generative sensing module will have to learn the codebook from warped inputs, which could inevitably encode missing holes into the codebook. On the other hand, by finetuning the encoder from a converged first-stage checkpoint, we consistently improve the performance of our model across all image level metrics. We refer the readers to the supp. materials for detailed numbers.

**Real-world scene:** We apply SGAM and the baselines to real-world images that we scraped from Google Earth (see Fig. 8), with global mapping visualized in Fig. 5. While our results produce less realistic results than InfiniteNature in one-step prediction (Tab. 4), InfiniteNature cannot estimate the geometry well in invisible regions, and thus compounding error occurs during unrolling as shown in Tab. 3. GFVS-explicit suffers from severe domain shift. Since the images contain more diverse and complicated scene structures, GFVS-implicit fails to capture the viewpoint shift precisely during unrolling. In contrast, SGAM can produce consistent and realistic results.

| Frame 1 | Frame 10 | Frame 30 | Frame 50 | Frame 100 | Frame 200 | Frame 300 |

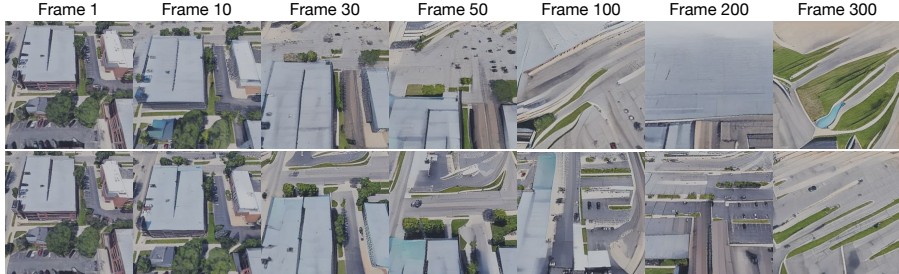

Figure 10: **Ablation Study on the effect of RGBD-integration** to SGAM on GoogleEarth-infinite dataset. Top: no RGBD-integration. Bottom: with RGBD-integration.

| Renderer | FID↓ |
|---|---|
| InfiniteNature ([40]) | 182.6 |
| GFVS-implicit ([61]) | 160.40 |
| GFVS-explicit ([61]) | 133.12 |
| Ours | **79.26** |

Table 3: **Comparison study on scene generation** between ours and baseline methods on the GoogleEarth-Infinite validation set. FID is computed based on the generated sequences unrolled for 60 steps against randomly sampled frames from the GoogleEarth-Infinite validation set.

| Renderer | PSNR↑ | SSIM↑ | LPIPS↓ | $\mathcal{L}_{\text{disp.}}$ (vis) ↓ | $\mathcal{L}_{\text{disp.}}$ (invis) ↓ | $\mathcal{L}_{\text{disp.}}$ ↓ | ref-FID↓ | FID↓ |
|---|---|---|---|---|---|---|---|---|
| InfiniteNature ([40]) | **24.78** | **0.880** | **0.172** | **0.017** | 0.041 | 0.020 | **12.61** | **37.92** ±0.20 |
| GFVS-implicit ([61]) | 19.77 | 0.473 | 0.398 | - | - | - | 23.61 | 38.70 ± 0.28 |
| GFVS-explicit ([61]) | 20.09 | 0.494 | 0.398 | - | - | - | 35.88 | 44.51 ± 0.26 |
| Ours | 23.07 | 0.609 | 0.304 | **0.017** | **0.022** | **0.018** | 22.88 | 42.29 ± 0.16 |

Table 4: **Comparison study on one-step prediction** between ours and baseline methods on the GoogleEarth-Infinite validation set. $\mathcal{L}_{\text{disp.}}$ (vis) denotes the disparity $\mathcal{L}_1$ loss in the visible region at the target pose. $\mathcal{L}_{\text{disp.}}$ (invis) denotes the disparity $\mathcal{L}_1$ loss in the invisible region at the target pose. The disparity is scaled between 0 and 1. We highlight **best**, and second best scores.

**Diversity:** SGAM allows us to create diverse 3D world by varying camera trajectories and by varying the step size. As shown in Fig. 7 and Fig. 9, both of which are very effective. `Greedy` samples the next target pose based on the L1 distance of the visible region of the warped views.

**Global mapping:** The explicit 3D mapping ensures global consistency during generation steps. Without global mapping, direct depth-based image warping ([40]) hurdles inference quality at the generative refinement stage. Importantly, it cannot ensure global consistency when having a loop as shown in Fig. 4. We demonstrate in Fig. 10 that RGBD-integration denoised mapping can achieve much more diverse scenarios than the one without, which scene degrades to large empty airport ground around 200 steps of unrolling becomes it unable to fit latent code to fit noisy features.

**Societal impact:** Realistic 3D generation technology has many applications, such as visual content creation and simulation. Unfortunately, it may also be used to spread misinformation.

# 5 Conclusion

In this paper, we present a novel 3D scene generation algorithm, SGAM, which create a consistent, realistic, and large-scale 3D virtual world through simultaneous generation and mapping. Our result demonstrates we can achieve similar (in synthetic data) or even more realistic (in real-world data) sensor observation than methods with Transformer-based latent code sampler during unrolling.

**Acknowledgements:** The authors thank Derek Hoiem and David Forsyth for their early feedback. We thank Vlas Zyrianov for proofreading the paper. The project is partially funded by the Amazon Research Award and Illinois Smart Transportation Initiative STII-21-07. We also thank Nvidia for the Academic Hardware Grant.

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
