# Supplementary Material
# SGAM: Building a Virtual 3D World through Simultaneous Generation and Mapping

**Yuan Shen**[1]    **Wei-Chiu Ma**[2]    **Shenlong Wang**[1]

[1]University of Illinois at Urbana-Champaign    [2]Massachusetts Institute of Technology

{yshen47, shenlong}@illinois.edu

weichium@mit.edu

## Abstract

In the supplementary material, we first introduce the architecture details for our method and other baselines in Sec. 1. We then provide a detailed description of the evaluation metrics in Sec. 2. Finally, we provide additional quantitative and qualitative results and analysis on GoogleEarth-Infinite and Clevr-Infinite datasets in Sec. 3. We strongly recommend the readers to check our project page at https://yshen47.github.io/sgam/, which gives a brief introduction to our approach and provides additional qualitative results in video sequences.

## 1 Experimental Details

### 1.1 Depth Map Representation

Following previous work (5; 2), we use the normalized inverse depth map to encode geometry in our RGB-D image representation. We normalize the scale of the inverse depth maps to be between -1 and 1 for each dataset: $d = \frac{d - d_{\min}}{d_{\max} - d_{\min}} \cdot 2 - 1$, where $d_{\min}, d_{\max}$ is computed from the training set for each dataset.

### 1.2 Mapping

To ensure a fair comparison of 3D scene generation, we augment all the existing baselines (5; 4) with the same mapping module as SGAM. Specifically, we leverage the volumetric integration approach described in ElasticFusion (7) as the mapping module for all competing algorithms. It is a scalable volumetric integration implementation based on the voxel-hashing scheme. The voxel length is 0.01, and the SDF truncation value is 0.05.

### 1.3 GoogleEarth-Infinite Dataset

To validate the effectiveness of SGAM on real-world data, we collect a posed real-world RGB-D dataset, GoogleEarth-Infinite, in which we extract 3d mesh from GoogleEarth using MapsModelsImporter API[*] and render the mesh with Blender. GoogleEarth-Infinite contains 350k RGB-D images at a resolution of 512x512, distributed across 30 training scenes, and 3 validation scenes. The scene meshes are sampled near residential areas around several US university campuses. For each scene, we sample poses in a dense grid parallel to the ground plane, such that there is a small extrapolation area if warping from nearby poses.

---

[*]Google Earth 3D mesh extraction API: https://github.com/eliemichel/MapsModelsImporter

36th Conference on Neural Information Processing Systems (NeurIPS 2022).

## 1.4 Implementation Details for SGAM

**Architecture** Our generative sensing model follows a similar convolutional encoder and decoder models as presented in VQGAN [2] with a few differences. Specifically, our encoder contains five convolutional residual blocks and the decoder contains five residual convolutional residual blocks. Each residual block consists of two stack of modules with Group-Norm+SwishNonLinearity+Conv2d+shortcut connection. Codebook vocabulary size $|Z|$ is 16384 for both datasets and codebook embedding dimension is 256. The total length of sequence or latent code length for one image on CLEVR-Infinite is 256, and 512 on GoogleEarth-Infinite. The output dimension of VQGAN and the input dimension of the discriminator, $C$, is 4, which corresponds to the RGB and disparity. In total, there are 100M trainable parameters.

**Training and inference details** During the second-stage training, we trained 180k iterations on CLEVR-Infinite, and 120k iterations on GoogleEarth-Infinite. During training, we random sample 1 or 2 sources poses nearby with camera center within 3 units in CLEVR-Infinite, within 0.15 units in GoogleEarth-Infinite.

## 1.5 Architecture Details for GFVS-Explicit

On CLEVR-Infinite, we follow similar architecture as is used in GFVS explicit-image variant in their codebase [5], except that we use one single RGB-D codebook to encode RGB and disparity. In total, there are 210M training parameters. However, on GoogleEarth-Infinite, due to an increase of image resolution to 512x512, we downscale the latent Transformer architecture such that it can fit into our GPU. Specifically, we keep 8 Transformer layers, i.e., $n_{layer}$, and 8 attention heads. The Transformer embedding size is 1024, and the block size is 2077. The sampling for source poses is the same as is used in SGAM.

## 1.6 Architecture Details for GFVS-Implicit

On CLEVR-Infinite, we follow similar architecture as is used in GFVS implicit-depth variant in their codebase [5] except that we use one single codebook to encode RGB-D without a separate depth codebook. The motivation for using a single RGB-D codebook as SGAM is to exclude the effect of codebook difference on performance comparison. In total, there are 303M training parameters. Similarly, as GFVS-Explicit, we downscale the latent Transformer architecture such that it can fit into our GPU. Specifically, we keep 8 Transformer layers, i.e., $n_{layer}$, and 8 attention heads. The Transformer embedding size is 1024, and the block size is 2077. The sampling for source poses is the same as is used in SGAM with 1 or 2 source poses during training. After average and max pooling separately over the last feature layers of Transformer across different source views, we concatenate the pooling results and feed them into an MLP (a linear layer from 2048 to 1024, ReLU, and a linear layer from 1024 to 16384(vocab_size)). It takes 120 iterations on CLEVR-Infinite, and 80k iterations until convergence on GoogleEarth-Infinite.

## 1.7 Architecture Details for InfiniteNature

We re-implement InfiniteNature [4] in Pytorch since the original TensorFlow implementation does not contain training scripts. We use the same hyperparameters based on their codebase and retrain their models. We also reproduce the render-refine-repeat perpetual view synthesis pipeline following their Tensorflow implementation. In total, there are 240M trainable parameters for InfiniteNature.

There is one major difference in our implementation. First, we remove the geometric grounding step since we find it harmful during inference in the CLEVR-Infinite dataset. Second, to reduce the domain shift issue on Google Earth, we only let the model fill in the invisible region in the target pose after forward warping pixels from the source view. Unlike other competing methods, InfiniteNature takes only one source image. For the CLEVR-Infinite dataset, we follow the same architecture as the original InfiniteNature codebase. With an increased resolution at 512x512 on GoogleEarth-Infinite, we add two additional convolution layers in the encoder; the channel sizes for the eight convolution output dimensions are [64, 128, 256, 512, 1024, 2048, 2048, 2048].

## 2   Evaluation Metrics

**Image-level metrics:**   we use scikit-learn implementation for PSNR and SSIM[†]. For LPIPS, we use the official implementation[‡]. For FID, we use the official Pytorch implementation[§].

**Scene-level measures:**   For the scene-level metric, we first sample two sets of point cloud blocks separately from the generated scenes, denoted as $\mathcal{S}_g$ and ground-truth scenes, denoted as $\mathcal{S}_r$. We divided the generated large-scale scenes into smaller blocks for all the competing methods and the ground-truth reference data to evaluate scene-level metrics. In total, we sample 1100 blocks from each set for evaluation. Following prior work ([1; 6]), we use Jensen-Shannon Divergence (JSD), Minimum Distance (MinDst), 1-nearest neighbor accuracy (1-NNA), and Coverage (Cov). We follow notations and equations used in PointFlow ([6]). In addition, we also report the Maximum Mean Discrepancy (MMD) ([3]), a non-parametric distance between two distributions.

**Kernel and distance functions:**   All the metrics above are based on computing either a pairwise distance function or a kernel function between two point clouds. To measure the distance/similarity robustly and efficiently, we convert each scattered point cloud block into a normalized volume-based 3D histogram (at the resolution of $10^3$, $15^3$, $20^3$). We then use the cosine similarity $S_C$ as the kernel function during kernel-based metric evaluation, such as MMD and JSD:

$$K(\mathbf{X}, \mathbf{Y}) = \frac{\mathbf{X} \cdot \mathbf{Y}}{||\mathbf{X}||\, ||\mathbf{Y}||},$$

where $\mathbf{X}$ and $\mathbf{Y}$ are normalized 3D histograms.

We also use cosine distance $D_C$ for distance-based metrics, such as MinDist and 1-NNA:

$$D(\mathbf{X}, \mathbf{Y}) = 1 - K(\mathbf{X}, \mathbf{Y}) = 1 - \frac{\mathbf{X} \cdot \mathbf{Y}}{||\mathbf{X}||\, ||\mathbf{Y}||}$$

Next, we will describe each metric in detail.

**Jensen-Shannon Divergence (JSD)** operates on discretized Euclidean 3D space, which measures the divergence between the marginal distribution of the generated and reference point clouds. Mathematically

$$\mathrm{JSD}(P_g, P_r) = \frac{1}{2} D_{KL}(P_r || M) + \frac{1}{2} D_{KL}(P_g || M),$$

where $P_r$ and $P_g$ are empirical distributions computed in the form of 3D histograms; $M = \frac{1}{2}(P_r + P_g)$; $D_{KL}$ is the KL divergence. However, JSD can be misleading in cases when the model always outputs an average shape point cloud that is close to $M$.

**Coverage (COV)** quantifies how many reference point clouds can be matched to at least one generated point cloud. COV can be used to detect mode collapse, even though it does not dig into the quality of generated point clouds. Mathematically,

$$\mathrm{COV}(\mathcal{S}_g, \mathcal{S}_r) = \frac{|\{\arg\min_{\mathbf{Y} \in \mathcal{S}_r} D(\mathbf{X}, \mathbf{Y}) | \mathbf{X} \in \mathcal{S}_g\}|}{|\mathcal{S}_r|}$$

**Minimum matching distance (MinDst)** measures the distance of each generated point cloud to its nearest neighbour in the reference dataset:

$$\mathrm{MinDst}(\mathcal{S}_g, \mathcal{S}_r) = \frac{1}{|\mathcal{S}_r|} \sum_{\mathbf{Y} \in \mathcal{S}_r} \min_{\mathbf{X} \in \mathcal{S}_g} D(\mathbf{X}, \mathbf{Y})$$

MinDst serves to measure the quality of generated point cloud, but cannot reflect well low-quality point clouds in $\mathcal{S}_g$, since only the point cloud with minimum distance to the reference is considered in this metric.

---

[†]https://scikit-image.org/docs/stable/api/skimage.metrics.html
[‡]LPIPS: https://github.com/richzhang/PerceptualSimilarity
[§]FID: https://github.com/mseitzer/pytorch-fid

| Model | 3D Hist. Res. | JSD $(10^{-2})\downarrow$ | MMD $(10^{-5})\downarrow$ | MinDist $\downarrow$ | 1-NNA $\downarrow$ | Cov $\uparrow$ |
|---|---|---|---|---|---|---|
| GFVS-implicit [5] | | 0.600 | 13.11 | **0.138** | 0.700 | 0.427 |
| GFVS-explicit [5] | $10^3$ | 8.838 | 372.900 | 0.150 | 0.874 | 0.170 |
| Ours | | **0.280** | **9.890** | 0.143 | **0.615** | **0.533** |
| GFVS-implicit [5] | | 0.775 | 4.510 | **0.205** | 0.675 | 0.376 |
| GFVS-explicit [5] | $15^3$ | 10.870 | 211.500 | 0.215 | 0.825 | 0.122 |
| Ours | | **0.656** | **4.441** | 0.221 | **0.585** | **0.454** |
| GFVS-implicit [5] | | **0.994** | **2.776** | **0.269** | 0.695 | 0.303 |
| GFVS-explicit [5] | $20^3$ | 11.660 | 66.610 | 0.270 | 0.775 | 0.116 |
| Ours | | 1.26 | 2.967 | 0.294 | **0.566** | **0.360** |

Table 1: **Scene-level comparison study** on generated 30x30 scenes with prior arts on CLEVR-Infinite. Compared with Table 2 in the main paper, we provide results on two other resolutions.

**Maximum Mean Discrepancy (MMD)** estimates two distributions by comparing the mean squared differences of the statistics from two sample sets. Mathematically,

$$\text{MMD}(\mathcal{S}_g, \mathcal{S}_r) = \frac{1}{|\mathcal{S}_r|^2} \sum_{\mathbf{X}\in\mathcal{S}_r} \sum_{\mathbf{X}'\in\mathbf{S}_r} K(\mathbf{X}, \mathbf{X}') - \frac{2}{|\mathcal{S}_r||\mathcal{S}_g|} \sum_{\mathbf{X}\in\mathcal{S}_r} \sum_{\mathbf{Y}\in\mathcal{S}_g} K(\mathbf{X}, \mathbf{Y}) + \frac{1}{|S_g|^2} \sum_{\mathbf{Y}\in S_g} \sum_{\mathbf{Y}\in\mathcal{S}_g} K(\mathbf{Y}, \mathbf{Y}')$$

Leveraging the kernel tricks, MMD can measure the distribution similarity from all the moments.

**1-nearest neighbor accuracy (1-NNA)** aims to decide if two samples of point clouds are from the same distribution. It achieves this purpose by classification with a 1-NN classifier. If the 1-NN classifier has high accuracy, then two samples are easy to tell apart and thus are from different distributions, and vice versa. Mathematically,

$$\text{1-NNA}(\mathcal{S}_g, \mathcal{S}_r) = \frac{\sum_{\mathbf{X}\in\mathcal{S}_g} \mathbb{1}[N_{\mathbf{X}} \in \mathcal{S}_g] + \sum_{\mathbf{Y}\in\mathcal{S}_r} \mathbb{1}[N_{\mathbf{Y}} \in \mathcal{S}_r]}{|\mathcal{S}_g| + |\mathcal{S}_r|},$$

where $N_{\mathbf{X}}$ is the nearest neighbor of X among the union of $\mathcal{S}_r$ and $\mathcal{S}_g$ excluding $\mathbf{X}$, and $N_{\mathbf{Y}}$ is defined in a similar formulation.

# 3  Additional Results

## 3.1  CLEVR-Infinite Scene-level Metrics at Different Resolutions

We provide scene-level evaluation with 3D histogram resolution at $10^3$, $15^3$, and $20^3$ in Table 1. Ours is consistently the best for 1-NNA and Cov at different resolutions, which suggests our prediction matches with the scene-level distribution of the ground-truth distribution. At coarser resolution, we achieve the best for JSD and MMD, but achieve the second-best at the lowest resolution. Finally, for MinDist, we notice that it does not reflect the scene generation performance well. As is shown in Figure 3 of the main paper, despite a worse quality of GFVS-explicit at scene-level, GFVS-explicit achieves a higher score than ours at lower resolutions, indicating the lower quality generated point clouds are not captured by MinDist.

## 3.2  Diversity

**Effects of temperature sampling**   In Figure 2, we provide qualitative uncertainty on temperature sampling. We visualize both uncertainties in latent code space and the color space. For uncertainty quantification in latent code space, we visualize the entropy of each latent code. To quantify the image-level uncertainty, we sample 100 images using temperature sampling and visualize the standard deviation for all the RGB samples at the pixel level. This figure shows that outpainting regions and geometry boundaries tend to have higher color-space uncertainty. Nevertheless, uncertainty in latent code is not coupled with appearance uncertainty. We conjecture that our codebook has certain redundancies.

**Effects of the number of source views** We also compare different image-based rendering strategies in Fig. 1. As shown in the figure, the generated scene's diversity increases if fewer source views are used during the image-based rendering stage. Nevertheless, the appearance consistency across time decreases if we leverage fewer source images for inverse warping. This result suggests that finding a good trade-off between consistency and diversity is crucial for high-quality scene generation in SGAM.

### 3.3 Failure cases on GoogleEarth-Infinite

We collect several typical failure cases of SGAM in Figure 3. Despite producing realistic footage and 3D scene in most cases, SGAM tends to expand the building footprints to an unrealistic scale during the recursive sampling (row 1 and row 2). Additionally, SGAM outpaints better along the vertical direction than moving horizontally on the GoogleEarth dataset (row 3) since the same camera moving step results in a larger outpainting gap if moving horizontally than moving forward. We anticipate this could be resolved through a minor relative motion step. Finally, compared with natural objects, e.g., trees, SGAM is likely to struggle to generate complex structures (row 4). Training a stronger-capacity codebook might alleviate this issue.

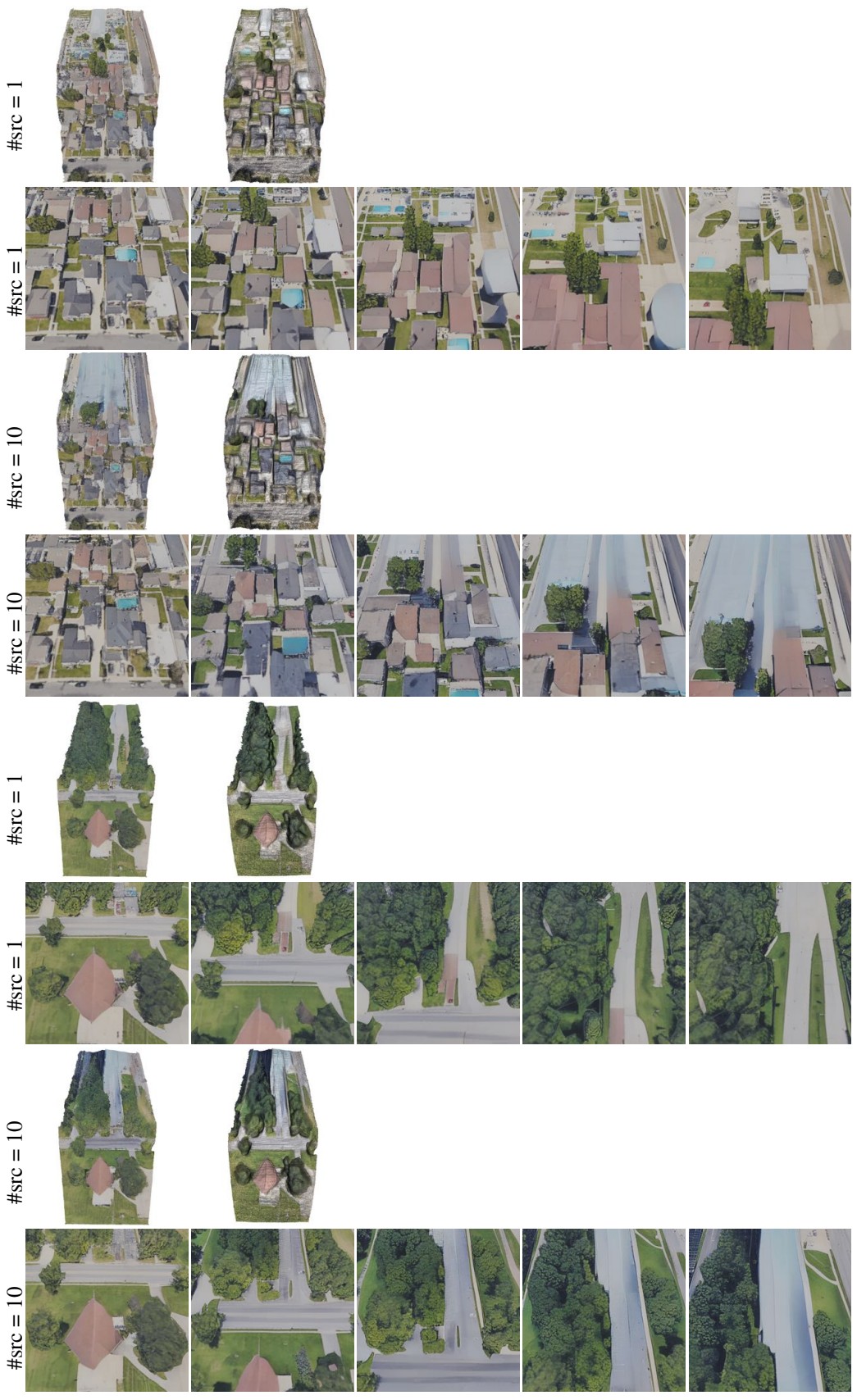

Figure 1: **Effect of the source view number on diversity**. The rows in odd numbers consist of birds-eye view in the form of color point clouds and RGB-D integrated global meshes. Each sequence unrolls for 70 steps with the same relative movements between frames

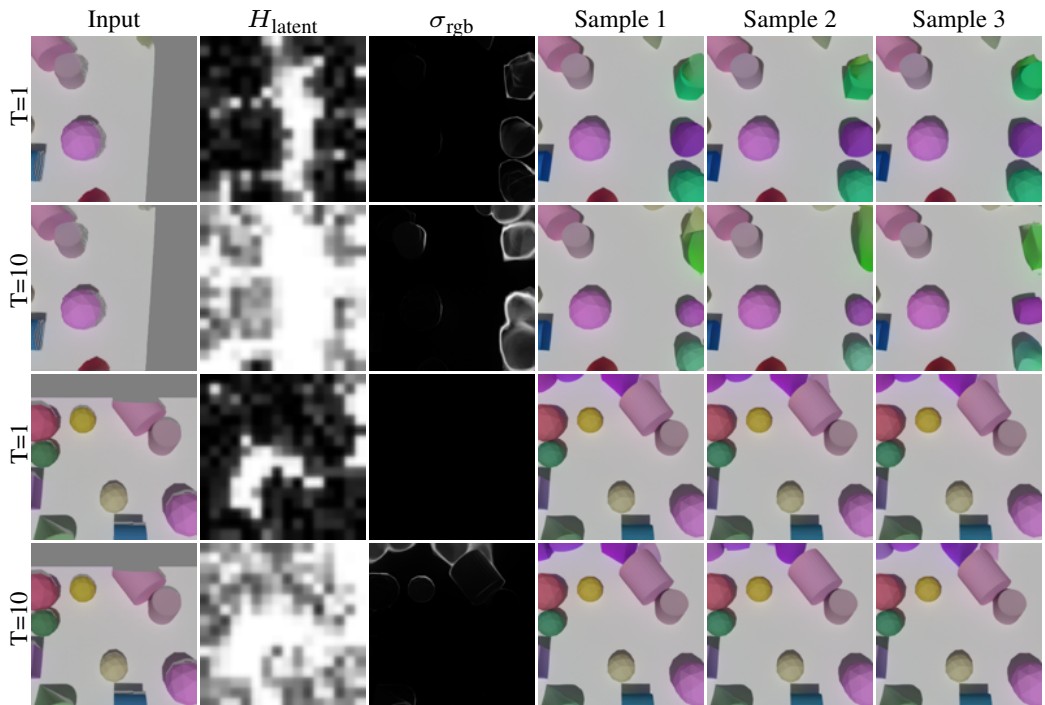

Figure 2: **Diversity samples using different temperature.** $H_{\text{latent}}$ denotes the entropy of latent code distribution. $\sigma_{\text{rgb}}$ is the per-pixel standard deviation based on 100 samples from temperature sampling. T is the temperature parameter.

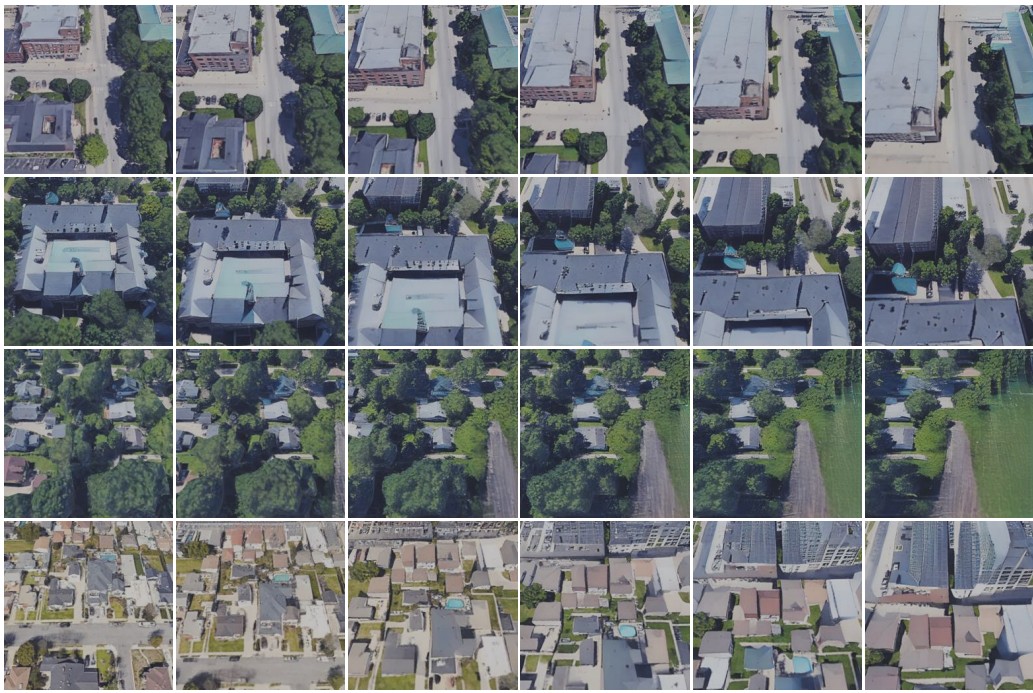

Figure 3: **Failure Cases of SGAM on GoogleEarth-Infinite**