# OpenReview forum: "SGAM: Building a Virtual 3D World through Simultaneous Generation and Mapping"
_NeurIPS.cc/2022/Conference — NeurIPS 2022 Accept_

### Official Review · Reviewer_GmLE · 2022-07-07

**Rating:** 3
**Confidence:** 5
**Soundness:** 3 good
**Presentation:** 3 good
**Contribution:** 1 poor

**Summary:**

The paper proposes a method to generate a large-scale RGBD image in a near NADIR view. It uses a standard image generative model VQGAN while using the already-generated scenes as an input constraint. It qualitatively and quantitatively evaluate on the Clever dataset. For Google earth data, only qualitative evaluations are given.

**Questions:**

I want answers to my criticism (weakness analysis) above.

**Limitations:**

The limitation description is OK. I cannot think of a better one.

**Strengths And Weaknesses:**

To me, this paper is over-selling. The paper is just a 4 channel image generation by repeatedly applying a standard VQGAN. Simultaneous generation and mapping does not make much sense.

The core of the method is an application of an existing technique and the technical contribution is weak. I also do not like the way authors present their method to handle a general camera pose, while in reality they just use near nadir views.

Also, CLEVER scene is a bit too simple and through evaluations should be given on real data (Google Earth).

---

> ### Author Response · Authors · 2022-08-02
> **Responses to Reviewer GmLE**
>
> **Simultaneous generation and mapping is critical:** We strongly believe SGAM is a critical and innovative step towards perpetual 3D scene generation.Through this paper, we also hope to convey the importance of explicitly 3D modeling in large-scale scene generation. While we indeed exploit VQ-GAN and KinectFusion in SGAM (*i.e.*, leverage KinectFusion for volumetric map building, adopt VQ-GAN for generative sensing, etc), *why they are used* and *how they are used* are all carefully designed. The resulting framework is generic, interpretable, and can be applied to various setup. It is not just a simple extension. Also, exploiting existing algorithms to realize a novel idea does not mean there is no technical contribution. We hope the reviewers, in particular **Reviewer GmLE**, can acknowledge this.
>
>
> **Repeatedly applying VQ-GANs will not work:** We want to highlight our task is a full 3D generation instead of perpetual 2D image generation on the image plane. We are **NOT** repeatedly applying VQ-GAN, which will not work for this task.
> We need at least two critical components to achieve our goal of simultaneously perpetual sensor and 3D shape generation: 1) render novel views at a new pose by reflecting the stereo-parallax. 2) building a 3D representation ensure consistency over the long run. Solving these two challenges is the core intellectual merit of this paper.
>
> **Experiments on real-world data:** We want to highlight that GoogleEarth-Infinite is a challenging real-world dataset that we collect from GoogleEarth. The FID scores shown below demonstrate our performance for long-term video sequence generation.
>
> FID scores on generated GoogleEarth-Infinite dataset (unrolling 60 frames)
> | InfiniteNature | GFVS-implicit | GFVS-explicit |Ours |
> | -------- | -------- | -------- |-------- |
> | 182.6     | 160.4     | 133.1     | **79.26** |
>
>
> **Camera poses beyond near nadir views:** The scenes shown in the paper are indeed generated with near nadir views. To demonstrate that SGAM is able to generate scenes from less structured trajectories and different camera viewpoints, we exploit two new setup: (i) we adopt a spiral trajectory and rotate the cameras along the yaw axis such that they always "look at" the center of the scene; (ii) we fixed the origin of the camera and rotate 360 degrees along its roll axis. Due to computational resource limits, we did not re-train our models with various camera angles. Instead, we simply adopt the model trained with fixed angles and perform scene generation. Surprisingly, despite such a domain gap, SGAM is still able to produce coherent, reasonable 3D scenes. We refer the readers to the revision for the qualitative results (see supp **Figure 2** ).
>
>
> In addition, following the reviewers suggestion, we are also running SGAM on the KITTI-360 dataset, an urban self-driving dataset from a first-person perspective. Due to time and resource limits, the model is currently training. Validation performance shows promising results in the generative sensing model.

---

### Official Review · Reviewer_SFD9 · 2022-07-11

**Rating:** 8
**Confidence:** 4
**Soundness:** 4 excellent
**Presentation:** 3 good
**Contribution:** 4 excellent

**Summary:**

This paper presents a large scale scene generation method which promotes global 3D scene consistency. The method works by taking as input a random view of a scene, and then moving through the scene continually using the current understanding/representation to create incomplete observations from new perspectives, using a generative model to in-paint the incomplete regions, and then using the newly generated information to update the understanding/representation of the scene. With this setup they demonstrate impressive scene generations on a dataset made up of simples shapes scattered around a white room, and on google street view images.

**Questions:**

How difficult is this process to train? Have you observed consistent convergence to attractive scenes over multiple seeds? Does this green color bias effect all seeds similarly?

If I understand correctly the camera angle is fixed throughout the trajectory? Have you explored at all with random trajectories through the space with the camera orientation changing as you move? I would be interested how the method performed with less structured passes through the search space.

**Limitations:**

Yes

**Strengths And Weaknesses:**

This paper is interesting and the results are impressive. The method is strong, and integrates lots of different techniques from different areas to produce a novel and successful solution to a hard problem.

With respect to the method I am slightly skeptical how well it will scale to more complex scenes. The underlying 3D representation is a voxel grid, and while hashing is performed to make it more efficient, I am not sure if this will support complex detail that natural scenes poses. It might be good to speak to this in the limitations section.

Connected to this I comment I would have also liked to have seen applications of this method to complex indoor scenes, as shown in the GFVS paper. I realize you require depth which is not available in general for indoor image datasets, however there exists some simulated indoor datasets for which you could again capture depth in blender. This would really help to understand how well your model scales, as at the moment you only demonstrate performance on practically 2D maps.

---

> ### Author Response · Authors · 2022-08-02
> **Responses to Reviewer SFD9**
>
> **Limitation on voxel scene representations:** We agree that the resolution of the voxels will impact how much fine-grained details we can capture. It is a hyper-parameter that needs to be determined. The voxel-hashing we used for mapping is a popular strategy for scaling up volumetric representation at high precision. One can potentially adopt other special data structures such as Octotree to reduce memory usage and computational cost, besides voxel hashing. Thanks for the suggestions! We will include relevant discussion in the final version.
>
>
> **Results on complex scenes:** To validate whether our method can handle more challenging scenarios and first-person perspectives, we train SGAM on KITTI-360 dataset. KITTI-360 is a challenging self-driving dataset. It contains rigid objects (*e.g.,* cars), fine-grained geometry (*e.g., trees), reflections, and most importantly noisy sensory data. This allows us to evaluate how robust SGAM is. Our model is currently still training.
> Based on the the intermediate checkpoint, we can achieve **PSNR 23.80, SSIM 0.838, LPIPS 0.122, FID 37.88** on one-step prediction. Some preliminary qualitative results are shown in the revision. Due to computational resource limits, we will include the results on indoor synthetic scenes in the final version. We thank R2 for the great suggestion. We choose KITTI-360 since it allows us to verify multiple aspects at the same time (*e.g.,* robustness to the depth measurement).
>
>
> **On the convergence of training:** As mentioned in Sec. 3.4 and Sec. 4.3, we adopt a two-stage training strategy. Each stage is crucial. Missing either stage could have a detrimental effect. The color biases in CLVER arises from the environmental map that we used, rather than the random seed. We will change the environmental map and re-train our model to mitigate the issue.
>
> **Generating scenes from less structured trajectories:** The scenes shown in the paper are indeed generated with fixed camera angles. To demonstrate that SGAM is able to generate scenes from less structured trajectories and different camera viewpoints, we exploit two new setup: (i) we adopt a spiral trajectory and rotate the cameras along the yaw axis such that they always "look at" the center of the scene; (ii) we fixed the origin of the camera and rotate 360 degrees along its pitch axis. Due to computational resource limits, we did not re-train our models with various camera angles. Instead, we simply adopt the model trained with fixed angles and perform scene generation. Surprisingly, despite such a domain gap, SGAM is still able to produce coherent, reasonable 3D scenes. We refer the readers to the revision for the qualitative results (see supp **Figure 2** ).

---

### Official Review · Reviewer_QFY4 · 2022-07-11

**Rating:** 5
**Confidence:** 3
**Soundness:** 3 good
**Presentation:** 3 good
**Contribution:** 3 good

**Summary:**

The work tackles the problem of 3D scene generation, learning from sequences of RGB-D  images and its poses. The main contribution of the work comes from proposing an algorithm that uses a generative sensing module and a mapping module. The proposed method was evaluated on CLEVR and GoogleEarth dataset and achieves SOTA results on the former. The method is also more efficient compared to previous results.


**Questions:**

- In the video there seems to be flickers in GoogleEarth dataset around 1:54, where the scene changes abruptly? Could the authors explain why this happens? Or is this just another video and not a continuous scene change.


**Limitations:**

Yes, the authors addressed the limitations of the work. But I recommend adding the comments regarding the method requires exact depth information.

**Strengths And Weaknesses:**

Strengths:

- The proposed method SGAM uses generative sensing module and mapping module, where the former is a VQGAN and the latter is mostly KinectFusion using volumetric representations. The formulation is a sound integration of the neural network with previous methods.

- The paper is well written and easy to understand

- The proposed method achieves SOTA results on CLEVR dataset on image based metrics and also on 3D generative metrics (MMD, JSD, 1-NNA…).

- I enjoyed watching the results in video. I appreciate the authors for the hard work.


Weaknesses:

- My main concern lies in the experimental section, especially on real-world datasets. The main experiment was mainly on the CLEVR dataset. As the authors mentioned, the essence of scene generation comes from generating diverse but realistic results. Although the CLEVR dataset is a good way to show the distributional similarity between the generated set and the real distribution, the main limitation is that this is a synthetic dataset. The appendix in Table 4 shows that the method shows worse results on real dataset. I would like to hear from the authors regarding the discrepancy on the evaluation metrics on real and synthetic datasets.

- Compared to other methods [1, 2], the method requires accurate depth. What happens if the method is trained on ACID dataset, where the accurate depth information is unavailable? I regard this as an important problem, since many of the real-world applications do not provide the exact depth information.


Summary:
The proposed method that leverages generative sensing module and a mapping module makes sense. However, my main concern lies in the experimental section, where the method achieves best results only on synthetic dataset and requires accurate depth. I would like to hear the response from authors regarding the experimental section and listen to other reviewers before making the final decision.

===============================================================================

After rebuttal, I'm convinced that the proposed method has the strength of generating long-term, globally consistent scenes. Therefore I'm willing to increase the score. However, I strongly encourage the authors to compare the method against previous methods such as ACID and RealEstate10k datasets to show the strength/weakness in real/standardized datasets. I do not think achieving SOTA results on these datasets is crucial since the method has clear strength, but the method should at least be competitive to verify that the method works on real-world datasets.

---

> ### Author Response · Authors · 2022-08-02
> **Responses to Reviewer QFY4**
>
> **Robustness to depth measurement:** Our approach *does not* require accurate depth, even during training. In fact, the "GT depth" of our GoogleEarth-Infinite dataset contains noise. The "GT depth" is obtained by rasterizing the coarse meshes (which are built from SfM and MVS point clouds) that we crawled with the Google API. Therefore by nature it is merely a proxy geometry of the real world and is far from accurate. Fortunately, with the help of the VQ-based generative sensing module, we can learn to de-noise with the quantized codebooks and produce perpetual 3D scene without drifting. We also note that we use the same "GT depth" to train all baselines as well as our method. To further showcase the robustness of our approach, we are conducting experiments on KITTI-360. Specifically, we use the stereo estimation from deep nets to serve as the "GT" to train our model. Due to computational constraints, the training is still in progress. We, however, still provide some promising preliminary results in the revision for the one-step prediction results. Please see the response to Reviewer SFD9 for more details on KITTI.
>
> **Results on real world dataset:** We stress that our goal is to enable large-scale, long-term, globally consistent, perpetual 3D scene generation. Our method can generate at a much larger scale without domain drift. To verify our claim, we compute the FID score across all competing on GoogleEarth-Infinite by unrolling the perpetual generation for 60 steps with different initial images. All the methods follow the same predefined trajectory. Results show that our method significantly outperforms other methods:
>
> FID scores on generated GoogleEarth-Infinite dataset (unrolling 60 frames)
> | InfiniteNature | GFVS-implicit | GFVS-explicit |Ours |
> | -------- | -------- | -------- |-------- |
> | 182.6     | 160.4     | 133.1     | **79.26** |
>
> We also would like to highlight the qualitatitive results shown in Fig. 7.
>
> We also note that InfiniteNature predicts the next frame by directly warping pixels and refining the results. In the short term, such an inpainting-like scheme looks slightly faithful (since it aims to fill the residual) compared to the images generated from the discrete bottleneck, resulting in superior one-step prediction performance. However, we find the artifacts will thus accumulate over time and results in severe drifting compared to GFVS and ours. In contrast, ours uses a vector quantization in latent space, inducing strong prior and constraints. This expressiveness vs. prior trade-off makes our approach slightly worse than Infinitenature in one-step prediction (PSNR: 23 vs. 24, higher is better). Nevertheless, this discrete bottleneck helps prevent SGAM from domain drift, resulting in significantly better results long-term image generation quality (FID: 79 vs. 182, lower is better).
>
>
>
> **Flickering in videos:** Thanks for pointing this out! It is simply a visualization bug when we generate the video. The actual generation is continuous and the produced scene changes smoothly and coherently.

---

> > ### Comment · Reviewer_QFY4 · 2022-08-05
> > **Further questions**
> >
> > I appreciate the authors for the detailed response.
> >
> > I may have been confused with line 216 "Unfortunately, existing static, large-scale 3D datasets, such as ACID [40] and RealEstate10K [84], do not provide accurate depth.". Could the authors ellaborate the meaning of this sentence? I thought this was to justify the reason of creating the CLVEVR-Infinite dataset, which other real-world datasets (ACID, RealEstate10k) do not provide. I acknowledge that CLEVR-Infinite seems to be a great way of computing the generative metrics because we can obtain the ground truth geometry, but the authors mentioned "accurate depth" instead of "ground truth geometry", which confuses me.
> >
> > Also, if no exact depth is required, could the authors also explain why the proposed method was not evaluated on either ACID or RealEstate 10k dataset? These seem to be the standardized dataset for evaluating 3D scene generation tasks. If the answer is to focus on "large-scale, long-term, globally consistent generation", I do not think that achieving SOTA results on these datasets are crucial. However, for future works, I think adding the results can improve the paper by showing. This may be the weakness of the proposed method, but as long as the mentioned strength exists clearly, I don't think it will hurt the contribution of this work.

---

> > > ### Author Response · Authors · 2022-08-05
> > > **Thank you for the follow-up questions!**
> > >
> > > We thank Reviewer QFY4 for the prompt response! Please see the clarifications below.
> > >
> > > **Wordings:** We agree with Reviewer QFY4 that the wording is confusing. We will change from "*accurate depth*" to "*complete geometry*". Thanks for pointing it out!
> > >
> > > **Further clarifications on geometry requirement:** In this paper, we focus on the challenging task of *large-scale, long-term, globally consistent 3D scene generation*. In contrast to prior art where most of them targeted 2D image generation and only considered 2D metrics,
> > > we are interested in not only generating appealing, realistic appearances (in the form of 2D images), but also reconstructing a coherent 3D structure (in the form of 3D maps). Therefore, *in order to properly evaluate the effectiveness of our model*, both in terms of 2D and 3D metrics, we must rely on datasets that have complete geometry. Unfortunately, both ACID and RealEstate10K lack GT geometry, preventing us from assessing the quality of our 3D generations. We thus build two new large-scale, diverse dataset with GT geometry so that we can evaluate the 3D metrics objectively.
> > >
> > > **Extending SGAM to ACID/RealEstate10K:** Thanks for the suggestion! We agree with Reviewer QFY4 that applying SGAM to the two existing benchmarks is feasible. While both datasets do not provide GT geometry and are designed for 2D generative models and view extrapolation, we can still exploit muli-view stereo to compute proxy depth maps and use them to supervise SGAM. As discussed in the original rebuttal, SGAM is robust and is able to learn an effective codebook even when facing noise in geometry. We, however, note that even if we have trained such a model, we will still only be able to assess the perceptual quality of 2D RGB images on these datasets, which is only one of the two primary goals of this paper (the other is realistic, consistent and large-scale 3D scene generation).
> > > Due to resource constraints, we do have the capacity to start the training right now. We will try out best to include such results in the final version!

---

### Author Response · Authors · 2022-08-02
**Response to all reviewers**

We thank the reviewers for their insightful comments and valuable suggestions. We are very excited that the reviewers appreciated the novelty and soundness of our approach (*i.e.,* integrating deep generative models with prior art on mapping) [**Reviewer QFY4, Reviewer SFD9**], found the paper interesting and well-written [**Reviewer QFY4, Reviewer SFD9**], and acknowledged our extensive evaluation and impressive results on the large-scale synthetic dataset [**Reviewer QFY4, Reviewer SFD9**].

---

**Novelty and technical contributions**

As demonstrated in the paper (and further below), *simultaneous generation and mapping is the key to producing a large-scale, realistic, globally consistent 3D world*. Specifically, by grounding scene generation with mapping, one can generate a diverse set of scenes that *are coherent with* existing appearance and structure; it also allows one to reproduce mapped regions *with consistency*. Through iteratively updating the map, one can further expand the generation process to an extremely large scale *without drifting*.

We strongly believe SGAM is a critical and innovative step towards perpetual 3D scene generation.Through this paper, we also hope to convey the importance of explicitly 3D modeling in large-scale scene generation. While we indeed exploit VQ-GAN and KinectFusion in SGAM (*i.e.*, leverage KinectFusion for volumetric map building, adopt VQ-GAN for generative sensing, etc), *why they are used* and *how they are used* are all carefully designed. The resulting framework is generic, interpretable, and can be applied to various setup. It is not just a simple extension. Also, exploiting existing algorithms to realize a novel idea does not mean there is no technical contribution. We hope the reviewers, in particular **Reviewer GmLE**, can acknowledge this.

---
We now address the concerns of each reviewer individually. We have also included new experimental results per reviewers' request in the revision (**highlighted in blue**) and supplementary video. We strongly encourage the reviewers to take a look at **our revised supplementary material**.

---

### Meta-Review · Area_Chair_9WJz · 2022-08-27

**Recommendation:** Accept
**Confidence:** Certain

**Metareview:**

The paper explores generation of a volumetric (voxel map via hashing, a la KinectFusion et seq) from a sequence of 2D images.  This is achieved by synthesizing sensor images, and feeding them into a mapping module (like KinectFusion).  As the reviewers note, this is an interesting goal, and the approach is reasonable, and no novelty of the overall system was questioned.

However, the reviewers concur that the synthetic dataset is not sufficient to give confidence that the system is effective in practice.  The rather more limited Google Earth (GE) examples, do, however, provide evidence that this strategy is effective.

As a reader, the most important figure for me is Fig 3 in the supmat - a qualitative view of the rerendered GE scene.  It is quite clear that the rerendered scenes have learned the characteristic 3D structures of the GE datasets, indicating that the scene-specific training data is a strong contributor to the results.

The paper would have been stronger if such qualitative results had been shown for ACID scenes, either trained on Google maps (even if the training needed to be e.g. coastline specific), or trained on depth-from-stereo as indicated in the rebuttal.


**Award:**

No

---

### Decision · Program_Chairs · 2022-09-14

Accept